# Therapeutic Potential of Bromodomain and Extra-Terminal Domain Inhibitors for Synovial Sarcoma Cells

**DOI:** 10.3390/cancers16061125

**Published:** 2024-03-11

**Authors:** Yuki Kotani, Yoshinori Imura, Sho Nakai, Ryota Chijimatsu, Haruna Takami, Akitomo Inoue, Hirokazu Mae, Satoshi Takenaka, Hidetatsu Outani, Seiji Okada

**Affiliations:** 1Department of Orthopaedic Surgery, Osaka University Graduate School of Medicine, 2-2 Yamadaoka, Suita 565-0871, Japans.nakai.xrs@osaka-u.ac.jp (S.N.); seokada@ort.med.osaka-u.ac.jp (S.O.); 2Center for Comprehensive Genomic Medicine, Okayama University Hospital, Okayama 700-8558, Japan; rchijimatsu@okayama-u.ac.jp; 3Department of Orthopaedic Surgery, Osaka International Cancer Institute, Osaka 540-0008, Japan

**Keywords:** synovial sarcoma, BCL2 family, bromodomain and extra-terminal domain protein inhibitor

## Abstract

**Simple Summary:**

Synovial sarcoma, a rare type of soft-tissue sarcoma, currently lacks efficacious anticancer therapies. This sarcoma is characterized by the SS18-SSX fusion gene, which plays a pivotal role in epigenetic regulation. Our study centered on epigenetic modulators, particularly bromodomain and extra-terminal domain (BET) inhibitors. Bromodomains are “readers” of histone modifications that facilitate the initiation and elongation of gene transcription. BET inhibitors are known to obstruct bromodomain binding, thereby attenuating the expression of tumor-associated genes. This study demonstrates that BET inhibitors modulate cell-cycle regulators and members of the BCL2 family in synovial sarcoma, resulting in cell-cycle arrest and apoptosis. Furthermore, we observed that silencing SS18-SSX diminishes BCL2 expression and reduces sensitivity to BET inhibitors. Our findings indicate that BET inhibitors may effectively target the intrinsic apoptotic pathway modulated by SS18-SSX in synovial sarcoma, suggesting that BET inhibitors could represent a promising therapeutic approach for this malignancy.

**Abstract:**

Synovial sarcoma (SS), a rare subtype of soft-tissue sarcoma distinguished by expression of the fusion gene SS18-SSX, predominantly affects the extremities of young patients. Existing anticancer drugs have limited efficacy against this malignancy, necessitating the development of innovative therapeutic approaches. Given the established role of SS18-SSX in epigenetic regulation, we focused on bromodomain and extra-terminal domain protein (BET) inhibitors and epigenetic agents. Our investigation of the BET inhibitor ABBV-075 revealed its pronounced antitumor effects, inducing G1-phase cell-cycle arrest and apoptosis, in four SS cell lines. Notably, BET inhibitors exhibited regulatory control over crucial cell-cycle regulators, such as MYC, p21, CDK4, and CDK6. Additionally, RNA sequencing findings across the four cell lines revealed the significance of fluctuating BCL2 family protein expression during apoptotic induction. Notably, variations in the expression ratio of the anti-apoptotic factor BCLxL and the pro-apoptotic factor BIM may underlie susceptibility to ABBV-075. Additionally, knockdown of SS18-SSX, which upregulates BCL2, reduced the sensitivity to ABBV-075. These findings suggest the potential utility of BET inhibitors targeting the SS18-SSX-regulated intrinsic apoptotic pathway as a promising therapeutic strategy for SS.

## 1. Introduction

Synovial sarcoma (SS) is a rare, aggressively progressing malignant soft-tissue tumor found in adolescents and young adults and is characterized by the fusion gene SS18-SSX arising from a t(X;18)(p11.2;q11.2) translocation [1,2]. Furthermore, compared with other soft-tissue tumors, SS has minimal genetic mutations apart from the fusion gene [3]. The fusion oncoprotein SS18-SSX replaces SS18, a subunit of the BAF complex, and epigenetically and globally alters gene expression by modulating the BAF complex localization [4,5]. These indicate that SS has the capability to orchestrate the oncogenic program solely through the fusion gene SS18-SSX. Based on the assessment of resection margins and distant metastases, the primary treatment for SS is surgical resection followed by radiotherapy or chemotherapy [6]. Existing chemotherapy options like doxorubicin and ifosfamide are commonly employed; however, their efficacy is limited, and options available for advanced stages of the disease are scarce. Despite multimodal therapy implementation, disease-specific survival at 10 years is approximately 60% [7]. Therefore, the development of innovative therapeutic approaches for SS is highly anticipated.

Bromodomain and extra-terminal domain (BET) proteins, represented by BRD4, play crucial roles in initiating transcription and promoting transcriptional RNA elongation. BRD4 identifies and binds to acetylated histones, promoting transcription initiation by recruiting RNA polymerase II (RNA-polII) to the promoter regions of genes. Additionally, BRD4 interacts with positive transcription elongation factor b (p-TEFb) to enhance transcriptional RNA elongation by releasing RNA-polII, which is dormant in the promoter-proximal regions, for active transcription [8,9]. BET inhibitors competitively inhibit these functions by binding to acetylated histones. Three distinct classes regulate histone acetylation: “writers” represented by histone acetyltransferase, “erasers” represented by histone deacetylase (HDAC), and “readers” represented by BET [10]. Several studies on SS have focused on HDAC inhibitors targeting “writers” [11,12,13]. Recently, BET inhibitors targeting “readers” have attracted much attention, and several studies have explored BET inhibitors as potential anticancer agents. These inhibitors have been reported to inhibit BRD4 binding in super-enhancer regions, which play a pivotal role in the recruitment of RNA-polII to promoter regions, thereby diminishing the expression of MYC and other oncogenic genes [14]. This leads to the induction of cell-cycle arrest and apoptosis across various tumor cells [15,16,17,18]. However, reports on the anticancer activity of BET inhibitors against SS are absent.

Herein, we investigate the antitumor effects of BET inhibitors on SS as a potential new agent for treating SS using (+)-JQ1 and ABBV-075.

## 2. Materials and Methods

### 2.1. Cell Culture

Our laboratory established the human cell lines Aska-SS (SS; CVCL_6C43), Yamato-SS (SS; CVCL_6C44), and Asra-EPS (epithelioid sarcoma; CVCL_W949). The SS cell lines, HS-SY-II (CVCL_8719) and SYO-1 (CVCL_7146), were graciously provided by Dr. Toguchida (Kyoto University, Kyoto, Japan) and Dr. Ozaki (Okayama University, Okayama, Japan), respectively. The clear cell sarcoma cell lines, MP-CCS-SY (CVCL_0J33) and KAS, were graciously provided by Dr. Moritake (Miyazaki University, Miyazaki, Japan) and Dr. Nakamura (Japanese Foundation for Cancer Research, Tokyo, Japan). The epithelioid sarcoma cell line, NEPS (CVCL_IS66), was graciously provided by Dr. Kawashima (Niigata University, Niigata, Japan). We purchased the SU-CCS1 (clear cell sarcoma; CVCL_B470), VAESBJ (epithelioid sarcoma; CVCL_1785), EW-8 (Ewing sarcoma; V618), HT-1080 (fibrosarcoma; CVCL_0317), and SW872 (liposarcoma; CVCL_1730) cell lines from the American Type Culture Collection (Manassas, VA, USA). We obtained normal human dermal fibroblasts (NHDF) from Kurabo (Osaka, Japan). The accession numbers listed with each cell line are registered in the cell line database Cellosaurus (https://www.cellosaurus.org/index.html, accessed on 26 July 2022). All cell lines were cultured under standard conditions at 37 °C with 5% CO_2_ and 100% humidity in Dulbecco’s modified Eagle’s medium (DMEM; Nacalai Tesque, Tokyo, Japan) supplemented with 10% fetal bovine serum (FBS; Sigma-Aldrich, St. Louis, MO, USA), 1% antibiotics (penicillin 100 IU/mL, and 100 μg/mL streptomycin; Invitrogen, Carlsbad, CA, USA). All cell lines underwent authentication through morphological examination, genotyping by PCR, and analysis of growth characteristics. Prior to experimentation, these cell lines were confirmed to be negative for *Mycoplasma* contamination through the use of TaKaRa PCR Mycoplasma Detection Set (Takara Bio Inc., Shiga, Japan).

### 2.2. Compounds

ABBV-075 was obtained from Selleck Chemicals (Houston, TX, USA). The compound was solubilized in dimethyl sulfoxide (Sigma–Aldrich) before its addition to the cell cultures for subsequent examination, in accordance with the manufacturer’s instructions.

### 2.3. WST-8 Cell Proliferation Assay

The SS cells were seeded into 96-well plates at a density of 5 × 10^3^ cells/well in triplicates and incubated with the compound or vehicle control for 48 h. The cell proliferation rate was determined using the Cell Count Reagent SF (Nacalai Tesque). Absorbance at 450 and 690 nm (reference wavelengths) was recorded using a microplate reader, and the relative cell proliferation rate was calculated.

### 2.4. Flow Cytometry

The SS cells were seeded into 6-well plates at a density of 3–12 × 10^5^ cells/well and cultured for 24 h. Subsequently, the compound or vehicle control was administered. After 24 h of treatment at the designated concentration, the cells were harvested and stained with a propidium iodide (PI) solution (25 μg/mL PI, 0.03% NP-40, 0.02 mg/mL RNase A, and 0.1% sodium citrate) for 30 min at 20–23 °C. Cell-cycle analysis was performed using a BD FACSVerse flow cytometer (Becton Dickinson, Franklin Lakes, NJ, USA) and BD FACSuite (Becton Dickinson), following the manufacturer’s protocol.

### 2.5. Western Blotting

SS cells were seeded into 6-well plates at a density of 3 × 10^5^ cells/well and incubated with the compound or vehicle control at the designated concentrations for 24 h. For lysate preparation, the SS cells were initially washed with phosphate-buffered saline and subsequently lysed in radioimmunoprecipitation assay buffer (Thermo Fisher Scientific, Waltham, MA, USA) supplemented with a 1% protease/phosphatase inhibitor cocktail (Thermo Fisher Scientific). Protein concentrations were determined using bicinchoninic acid (Thermo Fisher Scientific) following the manufacturer’s protocol. Equivalent protein amounts were separated using 4–12% Bis-Tris gels (Life Technologies, Waltham, MA, USA) and transferred onto polyvinylidene difluoride membranes (Nippon Genetics, Tokyo, Japan). Membranes were incubated in Tris-buffered saline (TBS) with Tween 20 (TBS-T) containing 5% skimmed milk at 20–23 °C and subsequently incubated with primary antibodies (Appendix A) diluted in Can Get Signal Solution 1 (Toyobo, Osaka, Japan) at 4 °C overnight. They were subsequently incubated with secondary antibodies diluted in Can Get Signal Solution 2 (Toyobo) at 20–23 °C for 1 h. After washing with TBS-T, images were obtained using a ChemiDOC touch system (Bio-Rad, Hercules, CA, USA).

### 2.6. Quantitative Real-Time Polymerase Chain Reaction (qRT-PCR)

Total RNA was isolated using the RNeasy Mini Kit (Qiagen, Hilden, Germany) and reverse transcribed to cDNA using ReverTra Ace qPCR RT Master Mix (Toyobo). Gene expression analysis was performed using the StepOnePlus Real-Time PCR System (Applied Biosystems, Waltham, MA, USA) and SYBR Green Real-time PCR Master Mix (Toyobo). The expression levels of target genes were normalized to those of GAPDH. The relative expression was determined using the 2-ΔΔCt method. Appendix A lists the PCR primers (forward and reverse) used in this study.

### 2.7. Small Interfering RNA (siRNA) Transfection

SS cells were seeded in 6-well plates at a density of 3 × 10^5^ cells/well and cultured for 24 h. Subsequently, the cells were reverse transfected using Lipofectamine RNAiMax (Invitrogen) following the manufacturer’s protocol. For this procedure, 20 nmol/L of siRNAs targeting SS18-SSX1 were used for Aska-SS, Yamato-SS, and HS-SY-II, and SS18-SSX2 for SYO-1 (Appendix A). A non-targeting negative control siRNA (Thermo Fisher Scientific) was also included.

### 2.8. Total RNA Sequencing (RNA-seq)

Total RNA was isolated from the four SS cell lines treated with ABBV-075 or the vehicle control for 24 h using the RNeasy Mini Kit (Qiagen). RNA-seq libraries were prepared using the NEBNext Poly(A) mRNA Magnetic Isolation Module (New England Biolabs, Ipswich, MA, USA) and the NEBNext UltraTMII Directional RNA Library Prep Kit (New England Biolabs). The libraries were sequenced on an Illumina NovaSeq 6000 system in paired-end 2 × 150 bp cycle mode, in accordance with the standard protocol.

### 2.9. Analyzing the Differentially Expressed Genes (DEGs)

We determined the DEGs following compound administration using iDEP (http://bioinformatics.sdstate.edu/idep/, accessed on 30 April 2023) [19]. The adjusted *p*-value was set at <0.05, and the log2 (fold change (FC)) was set at >1. Appendix A shows the DEG data.

### 2.10. Kyoto Encyclopedia of Genes and Genomes (KEGG) Pathway Analysis

KEGG pathway analysis was performed using the Database for Annotation, Visualization, and Integrated Discovery v6.8 (https://david.ncifcrf.gov/, accessed on 14 May 2023) [20,21]. Analyses were conducted with the species set as *Homo sapiens*, the identifier set as the official gene symbol, the gene list set as the list type, and the remaining parameters set to the default values. The results of the KEGG pathway analysis were annotated using the KEGG pathway. Appendix A presents the KEGG pathway analysis data.

### 2.11. Subcutaneous Xenograft Assays

All animal experiments were conducted in accordance with the protocols approved by the Institutional Animal Care and Use Committee of the Osaka University Graduate School of Medicine. Five-week-old male BALB/c nu/nu mice (SLC, Shizuoka, Japan) were used in this study. A subcutaneous injection of 1 × 10^7^ SYO-1 cells was administered to the right side of the back of individual mice for subcutaneous xenograft assays. Tumor dimensions were measured at 3 d intervals using a caliper, and tumor volume was calculated using the following formula: Volume = Length × Width × Width/2. Treatment intervention commenced when all tumors became palpable. Before treatment, the mice were randomly allocated to vehicle control and ABBV-075 groups. ABBV-075 was administered once daily at a dose of 1 mg/kg via oral gavage. The vehicle control group received an equivalent volume of saline via oral gavage once daily. The xenograft tumor volume and mouse body weight were measured at 3 d intervals. Mice were continually monitored for marked adverse effects, and all tumors were excised and weighed at the end of the treatment phase.

### 2.12. Histology

The tumors were fixed in a 10% formalin-neutral buffer solution (Wako, Osaka, Japan) and subsequently embedded in paraffin using a Tissue-Tek VIP 6 AI (Sakura Finetek, Tokyo, Japan). Paraffin-embedded tumors were sectioned at a 3.5 µm thickness and stained with hematoxylin and eosin (HE).

### 2.13. Immunohistochemistry (IHC)

Paraffin-embedded tumor sections were hydrated by passing through xylene and graded ethanol solutions. Antigen retrieval was facilitated by incubation in citrate buffer (pH 6.0) at 95 °C for 30 min, followed by inactivation of endogenous peroxidase activity using 3% hydrogen peroxide. The sections were then treated with Blocking One Histo (Nacalai Tesque) for 20 min to prevent non-specific binding. Subsequently, the sections were primarily incubated with a 1:200 dilution of Ki-67 (Cell Signaling Technology, Danvers, MA, USA) for 60 min. Secondary incubation was performed using N-Histofine Simple Stain MAX PO (MULTI) (Nichirei Biosciences, Tokyo, Japan) for 60 min. The immunoreactivity was visualized using Simple Stain DAB (Nichirei Biosciences), and contrast was enhanced with hematoxylin staining. All IHC analyses were performed using an Aperio CS2 (Leica, Wetzler, Germany), and the staining intensities were compared at ×200 magnification.

### 2.14. Statistical Analysis

All data are shown as means ± standard deviations (SDs). Statistical differences were assessed using a two-tailed Student’s *t*-test. Significance was set at *p*-values < 0.05, and the figure legends indicate precise *p* values.

## 3. Results

### 3.1. BET Inhibitors Exhibit Antitumor Effects for SS In Vitro

We conducted a cell viability assay using four SS cell lines and a NHDF cell line to examine the antiproliferative effects of BET inhibitors on SS. (+)-JQ1 caused a dose-dependent reduction in the number of viable cells in all four SS cell lines (Figure 1A). SS cells demonstrated greater sensitivity to (+)-JQ1 than to NHDF, with 50% inhibitory concentration (IC50) values of 0.04–0.45 µM and 20.1 µM, respectively (Table 1). Additionally, ABBV-075 caused a dose-dependent reduction in the number of viable cells in all four SS cell lines (Figure 1B). SS cells demonstrated greater sensitivity to ABBV-075 than to NHDF, with IC50 values of 0.7–67 nM and 2140 nM, respectively (Table 2). Over the course of 96 h, ABBV-075 exhibited dose- and time-dependent reductions in all four SS cell lines compared to the vehicle control (Figure 1C). In all four SS cell lines following ABBV-075 treatment, cleaved caspase-3 was detected (Figure 1D, Appendix A). We conducted cell viability assays using nine cell lines derived from five different sarcomas to discern variations in susceptibility among various soft-tissue sarcomas (Figure 1E–G). Notably, all nine cell lines exhibited reduced sensitivity compared to the SS cell lines (Table 2). These findings revealed that ABBV-075 induced apoptosis and inhibited SS cell proliferation in vitro.

### 3.2. ABBV-075 Delays SS Growth In Vivo

Additionally, we evaluated the antitumor effects of ABBV-075 on SS cells in vivo. Daily ABBV-075 administration significantly reduced tumor volume in the xenograft model compared to the vehicle control (Figure 2A). Histological examination using HE staining revealed that the tumors in the vehicle control group exhibited high cellular density, whereas those in the ABBV-075-treated group exhibited substantial internal necrosis and low cellular density (Figure 2B). Immunohistochemical staining of tumors using a Ki-67 antibody revealed a higher density of cells positive for Ki-67 in the vehicle control-treated group than in the ABBV-075-treated group (Figure 2C,D). These findings reveal that ABBV-075 slows the cell cycle and inhibits SS tumor growth in vivo.

### 3.3. ABBV-075 Inhibits Cell-Cycle Checkpoints and Causes G1 Arrest

Cell-cycle analysis revealed that ABBV-075 treatment increased the cell proportion in the G0/G1 phase across all four SS cell lines, accompanied by an increase in the G1 fraction in two SS cell lines (Figure 3A). We investigated the mechanisms underlying G1 arrest by examining the expression of cell-cycle regulatory proteins. MYC expression decreased, whereas p21 expression increased in a dose-dependent manner in all four cell lines. Furthermore, CDK4 and CDK6 expression, involved in directly advancing the cell cycle from G1 to S phase, decreased. Additionally, the increase in p21 expression was independent of p53 expression (Figure 3B, Appendix A). Similarly, mRNA expression analysis revealed elevated CDKN1A levels and decreased CDK4 and CDK6 levels in all four cell lines (Appendix A). These results indicate that ABBV-075 downregulates MYC and causes G1 arrest by inhibiting the cell-cycle checkpoint between the G1 and S phases.

### 3.4. ABBV-075 Targets the BCL2 Family Protein to Induce Apoptosis

We investigated the mechanism of the antitumor effect by employing RNA-seq to compile complete expression profiles from four SS cell lines treated with the vehicle control or ABBV-075. The mean gene counts of the four cell lines in the vehicle control and ABBV-075 groups were compared, and genes with a fold change >2 were defined as genes with variable expression. The numbers of upregulated and downregulated genes were 1970 and 2224, respectively (Figure 4A). KEGG pathway analysis was performed to identify the pathways regulated by ABBV-075 in SS. KEGG pathway analysis for all 4194 genes revealed that the p53 signaling pathway and apoptosis were the pathways that were regulated by ABBV-075 and were related to antitumor effects (Figure 4B). The BCL2 family is involved in p53 signaling and apoptosis pathways, and a heatmap of the BCL2 family was generated using gene count data from RNA-seq (Figure 4C). We also conducted Western blotting and qRT-PCR to explore the changes in expression levels of BCL2 family members following ABBV-075 treatment and their correlation with susceptibility to ABBV-075. The anti-apoptotic protein BCLxL was downregulated, whereas the pro-apoptotic proteins BIM and PUMA were upregulated by ABBV-075 (Figure 4D, Appendix A). ABBV-075 treatment reduced the mRNA expression of the anti-apoptotic factor BCLxL while concurrently enhancing the mRNA expression of the pro-apoptotic factors BIM and PUMA (Figure 4E). The mRNA expression of BCLxL in each cell line was compared with that of BCL2, which revealed a pronounced increase in BCLxL expression in Aska-SS and SYO-1 (Appendix A). Subsequently, the ratios of the anti-apoptotic factor to the pro-apoptotic factors were calculated, which revealed a significant alteration in the BCLxL:BIM ratio across all four cell lines (Table 3). Among the four cell lines, Aska-SS and SYO-1 exhibited greater susceptibility to ABBV-075 (IC 50 < 10 nM) than Yamato-SS and HS-SY-II (IC 50 > 10 nM) (Table 2). The pro-apoptotic shift in the BCLxL:BIM ratio was greater in the susceptible Aska-SS and SYO-1 strains (Table 4).

### 3.5. SS18-SSX Affects SS Susceptibility to ABBV-075

Several epigenome-modulating agents affect fusion gene expression in sarcoma cells [12,22,23]. We investigated the effects of ABBV-075 on SS18-SSX. We formulated an siRNA targeting SS18-SSX and silenced the fusion gene to explore the influence of SS18-SSX expression on the susceptibility to ABBV-075 (Figure 5A). We calculated the relative proliferation of SS cells treated with 10 nM ABBV-075 for the si_Control and si_SS18-SSX groups, with 0 µM ABBV-075 as the baseline. In the si_Control group, the relative proliferation was 0.32, 0.69, 0.13, and 0.80 for Aska-SS, Yamato-SS, HS-SY-II, and SYO-1, respectively. Conversely, in the si_SS18-SSX group, the relative proliferation was 0.80, 0.80, 0.80, and 0.55 for Aska-SS, Yamato-SS, HS-SY-II, and SYO-1, respectively. A significant reduction in susceptibility upon silencing of SS18-SSX was observed across three cell lines, Aska-SS, Yamato-SS, and SYO-1 (Figure 5B). We investigated the gene expression controlled by SS18-SSX using the Genome Expression Omnibus (GEO) database provided by the National Center for Biotechnology Information (GSE108028) to identify the pathway controlled by SS18-SSX and related to susceptibility to BET inhibitors in SS [4]. We compared the mean gene counts of the three cell lines in the sh_Control and sh_SS18-SSX groups and defined genes with a fold change >4 as genes with variable expression. The numbers of upregulated genes and downregulated genes were 1039 and 462, respectively (Figure 5C). Subsequently, when genes whose expression was downregulated by SS18-SSX silencing were sorted from those with significant differences in expression levels, BCL2 was identified as the most significant factor (Appendix A). The heatmap generated from the RNA-seq gene count data exhibited minimal variation in the expression levels of other members of the BCL2 family (Figure 5D). SS18-SSX silencing decreased BCL2 protein expression (Figure 5E, Appendix A). These findings indicate that susceptibility to ABBV-075 was influenced by SS18-SSX expression and that the intrinsic apoptotic pathway, modulated by BCL2, whose expression was upregulated by SS18-SSX, could potentially impact susceptibility.

## 4. Discussion

The efficacy of chemotherapy for SS is limited [7,24,25], and the development of further effective therapies is desired. The role of the SS18-SSX fusion gene in SS is progressively becoming clear, with emerging evidence suggesting that the fusion gene modifies BAF complex localization, thereby exerting an influence on epigenetic processes [4,5]. Agents that directly modify the BAF complex are unavailable, except for the BRD9 inhibitor [26]. Inhibitors targeting EZH2, a subunit of the polycomb complex that coordinates chromatin remodeling in conjunction with the BAF complex, are available [27]; however, they have not been clinically effective in SS [28]. Therefore, we investigated other epigenome-modifying agents for treating SS. Although sporadic reports of HDAC and DNMT inhibitors as agents capable of epigenetic regulation are available [11,12,29], investigations into BET inhibitors for SS are lacking. BRD4, a representative BET family protein, orchestrates transcriptional activation by recruiting the transcription elongation factor P-TEFb to the transcription start site through interaction with histone–lysine acetylation via its bromodomain [30]. Furthermore, BRD4 engages with the RNA polymerase II complex and participates in transcriptional elongation [31]. Although (+)-JQ1, the first BET inhibitor, was introduced in 2010 and has been extensively employed in studies on BET inhibitors, it is not currently undergoing human clinical trials because of its short half-life [14]. (+)-JQ1 inhibits BRD4 binding at the super-enhancer region of MYC, leading to the downregulation of MYC expression and modulation of downstream gene expression [32]. I-BET 151 induced G1 phase cell-cycle arrest and apoptosis for mixed lineage leukemia (MLL), achieved by reducing the binding of BRD4 to the promoter regions of MYC, BCL2, and CDK6 [33]. ABBV-075 is an orally bioavailable pan-BETi with a core of pyrrolopyridone that exhibits exceptional potency against BRD2, BRD4, and BRDT, with inhibition constants (Ki) of 1, 1.6, and 2.2 nmol/L, respectively. ABBV-075 induces G1 cell-cycle arrest and activates the intrinsic apoptotic pathway by downregulating BCLxL in hematologic tumor cell lines [18]. Additionally, ABBV-075 downregulates MYC, CDK6, and BCLxL expression, upregulates p21 and BIM expression, and induces apoptosis in acute myeloid leukemia (AML) cells [16]. Regarding soft-tissue sarcomas, several studies have demonstrated antitumor activity of BET inhibitors using Ewing sarcoma, rhabdomyosarcoma, and clear cell sarcoma cells through mechanisms similar to those of hematological tumor cells [17,22,23].

In SS, SS18-SSX contributes to CCND1 expression and stabilization; consequently, the activated CCND1/CDK4/CDK6 axis is considered a promising therapeutic target [34,35]. Additionally, fusion gene-dependent BCL2 overexpression and its contribution to tumorigenesis have been reported, and BCL2-related anti-apoptotic pathways are also considered a potential therapeutic target for SS [12,36,37,38].

Herein, MYC, CDK4, and CDK6 downregulation and p21 upregulation were observed following ABBV-075 treatment, which is consistent with previous reports. Furthermore, we confirmed the downregulation of the anti-apoptotic factor BCLxL and the upregulation of the pro-apoptotic factors BIM and PUMA following ABBV-075 treatment. A BET inhibitor would be a promising treatment for SS because ABBV-075 inhibits both pathways activated in SS. Although we demonstrated that SS18-SSX silencing reduced the susceptibility of ABBV-075 and that the BCL2-related apoptotic pathway is a potential pathway affecting susceptibility, BCLxL is possibly more critical than BCL2 in ABBV-075 treatment for SS. These results are consistent with a previous report that the antitumor effect of BCL2 inhibitors was poor despite BCL2 overexpression in SS, whereas BCLxL inhibitors exhibited nanomolar efficacy [38]. The possibility that BCL2 may not be as important in maintaining the cellular health of SS cells and that BCLxL may be more easily poisoned than BCL2 has been discussed, and the BCL2 family, rather than BCL2 alone, is considered a viable therapeutic target [38]. Moreover, a previous report on SS documented apoptosis induction by downregulating BCLxL, rather than BCL2, among the factors governing the intrinsic apoptotic pathway [39]. In the present study, the mRNA expression of BCLxL was substantially higher than that of BCL2 in Aska-SS and SYO-1, which are both susceptible to ABBV-075. The findings of this study also indicate that BCLxL expression could serve as a potential biomarker for predicting the therapeutic effectiveness of ABBV-075.

ABBV-075 completed a phase 1 clinical trial in 2019 and its safety and pharmacokinetics were investigated [40]. ABBV-075 exhibited anti-leukemic efficacy in patients with relapsed/refractory AML, and its effectiveness was further potentiated when combined with the BCL2 selective inhibitor, venetoclax [41]. Clinical trials on SS, where ABBV-075 induces G1 phase cell-cycle arrest and apoptosis, as in AML, are eagerly anticipated.

## 5. Conclusions

The BET inhibitor used in this study exhibited antitumor activity against SS cell lines by suppressing cell-cycle modulators and inducing apoptosis by regulating the BCL2 family proteins. Given that BCL2 is controlled by SS18-SSX and that SS18-SSX affects ABBV-075 susceptibility, BET inhibitors appear to be promising therapeutic agents for SS.

## Figures and Tables

**Figure 1 cancers-16-01125-f001:**
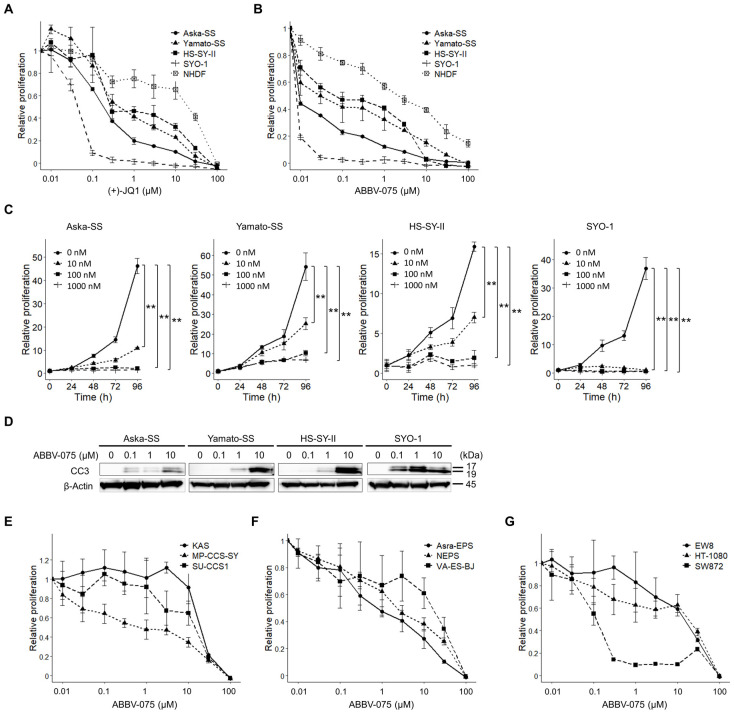
Antitumor effects of the bromodomain and extra-terminal domain protein (BET) inhibitors in vitro. (**A**) Relative cell proliferation measured using WST-8 assay of four synovial sarcoma (SS) cell lines and NHDF incubated with the different (+)-JQ1 concentrations for 48 h. (**B**) Relative cell proliferation measured using WST-8 assay of four SS cell lines and NHDF incubated with the different ABBV-075 concentrations for 48 h. (**C**) Relative cell proliferation measured using WST-8 assay of four SS cell lines incubated in the presence of ABBV-075 (0, 10, 100, 1000 (nM)) from 24 to 96 h. (**D**) Apoptosis marker expression of four SS cell lines treated with ABBV-075 via Western blotting. Relative cell proliferation measured using WST-8 assay of three clear cell sarcoma cell lines (**E**), three epithelioid sarcoma cell lines (**F**), and cell lines of Ewing sarcoma, fibrosarcoma, and liposarcoma (**G**). Data in (**A**–**C**,**E**–**G**) are means ± standard deviations. **, *p* < 0.01 (Student’s *t*-test).

**Figure 2 cancers-16-01125-f002:**
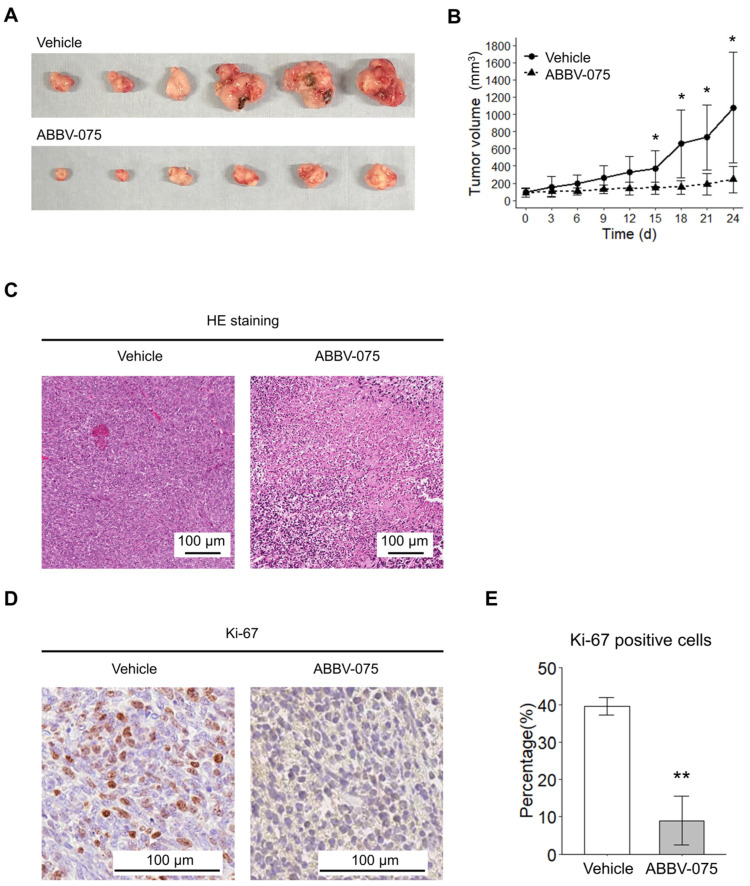
Antitumor effects of ABBV-075 in vivo. (**A**) Vehicle-treated tumors (above) and tumors treated with a 1 mg/kg concentration of ABBV-075 (below). (**B**) Mean tumor volume after treatment with the vehicle control and ABBV-075 (*n* = 6 mice in each group). (**C**) Hematoxylin-eosin staining of the excised tumor with the vehicle control and ABBV-075 at the end of the treatment phase. Ki-67 staining of the excised tumor with the vehicle control and ABBV-075 at the end of the treatment phase was performed using immunohistochemistry (**D**) and estimated as a percentage of positive cells (**E**). Data in (**B**,**D**) are means ± standard deviations. *, *p* < 0.05; **, *p* < 0.01 (Student’s *t*-test).

**Figure 3 cancers-16-01125-f003:**
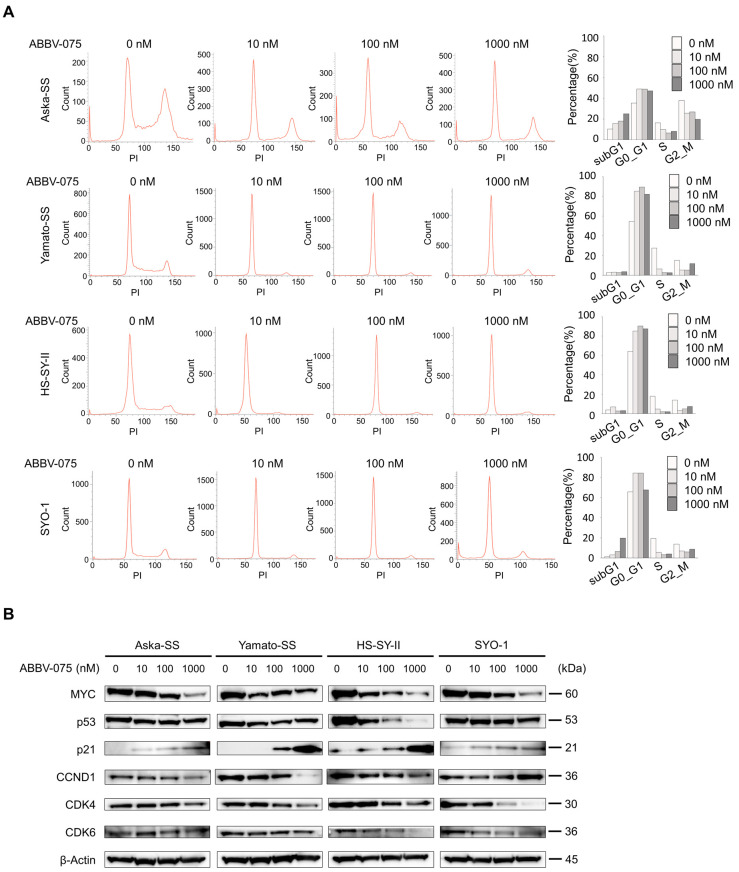
ABBV-075 induces G1 phase cell-cycle arrest. (**A**) Cell-cycle analysis of four SS cell lines treated with 0, 10, 100, and 1000 nM ABBV-075 was performed using flow cytometry. (**B**) Analysis of expression of cell-cycle regulatory proteins of four SS cell lines treated with 0, 10, 100, and 1000 nM ABBV-075 using Western blotting.

**Figure 4 cancers-16-01125-f004:**
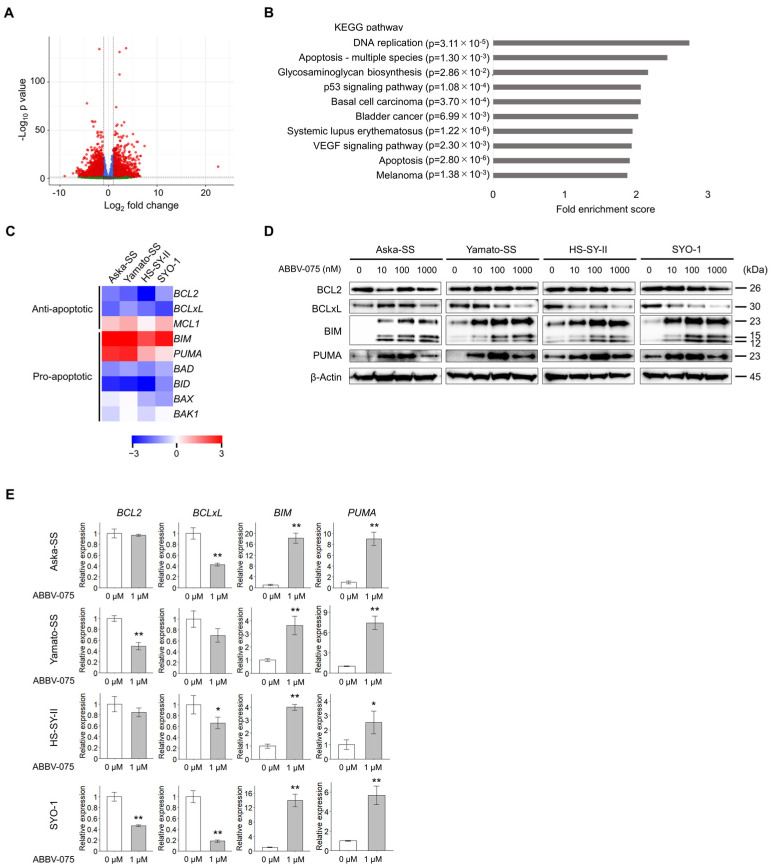
ABBV-075 targets BCL2 family. (**A**) A volcano plot of differentially expressed genes following ABBV-075 treatment compared with the vehicle control using mean gene counts of four synovial sarcoma (SS) cell lines. (**B**) Kyoto Encyclopedia of Genes and Genomes pathway analysis reveals the top 10 pathways regulated by ABBV-075 in the order of the fold enrichment score. (**C**) Heatmap of the BCL2 family expression based on gene count data obtained using RNA sequencing. (**D**) Expression of the BCL2 family proteins of four SS cell lines treated with 0, 10, 100, and 1000 nM of ABBV-075 analyzed by Western blotting. (**E**) mRNA expression of the BCL2 family of four SS cell lines treated with 0 µM and 1 µM ABBV-075 examined using quantitative polymerase chain reaction. Data in (**E**) are means ± standard deviations. *, *p* < 0.05; **, *p* < 0.01 (Student’s *t*-test).

**Figure 5 cancers-16-01125-f005:**
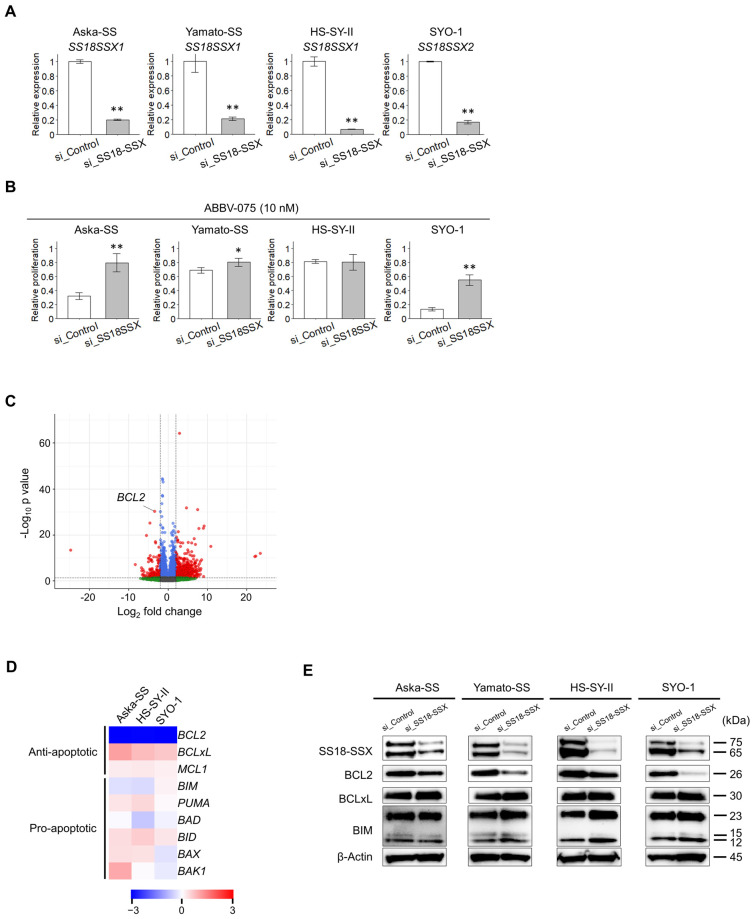
SS18-SSX regulates BCL2, modulating the intrinsic apoptotic pathway. (**A**) The silencing efficiency of SS18-SSX using small interfering RNA was evaluated using quantitative polymerase chain reaction. (**B**) SS18-SSX silencing reduces the susceptibility of SS cell lines to 10 nM ABBV-075 measured using the WST-8 assay. (**C**) A volcano plot of differentially expressed genes following SS18-SSX silencing compared with the control using mean gene counts of four synovial sarcoma (SS) cell lines. (**D**) Heatmap of the BCL2 family expression based on gene count data obtained using RNA sequencing. (**E**) Expression of BCL2 family proteins of four SS cell lines by silencing SS18-SSX compared with the control conducted using Western blotting. Data in (**A**,**B**) are means ± standard deviations. *, *p* < 0.05; **, *p* < 0.01 (Student’s *t*-test).

**Table 1 cancers-16-01125-t001:** IC50s of JQ1 for 4 synovial sarcoma cell lines and 1 human normal cell line.

	Cell Lines	IC 50 (µM)
Synovial sarcoma	Aska-SS	0.19
	Yamato-SS	0.45
	HS-SY-II	0.27
	SYO-1	0.04
Human dermal fibroblasts	NHDF	20.1

**Table 2 cancers-16-01125-t002:** IC50s of ABBV-075 for 14 cell lines from 6 human sarcomas and 1 human normal cell line.

	Cell Lines	IC 50 (µM)
Synovial sarcoma	Aska-SS	0.0048
	Yamato-SS	0.030
	HS-SY-II	0.067
	SYO-1	0.0007
Clear cell sarcoma	KAS	19.2
	MP-CCS-SY	0.67
	SU-CCS1	14.2
Epithelioid sarcoma	Asra-EPS	0.77
	NEPS	2.28
	VA-ES-BJ	15.8
Ewing sarcoma	EW8	14.5
Fibrosarcoma	HT-1080	17.9
Liposarcoma	SW872	0.11
Human dermal fibroblasts	NHDF	2.14

**Table 3 cancers-16-01125-t003:** Ratio of the anti-apoptotic factor to the pro-apoptotic factor.

Ratio	SYO-1		*p*-Value	Aska-SS		*p*-Value	Yamato-SS		*p*-Value	HS-SY-II		*p*-Value
	Vehicle	ABBV-075		Vehicle	ABBV-075		Vehicle	ABBV-075		Vehicle	ABBV-075	
*BCLxL*/*BIM*	15.36	0.20	0.001	13.82	0.32	0.002	9.79	1.92	2.02 × 10^−4^	6.02	1.00	7.07 × 10^−6^
*BCLxL*/*PUMA*	602.8	20.05	0.002	1029.4	45.80	ns	710.3	66.41	0.01	375.7	96.89	ns

**Table 4 cancers-16-01125-t004:** Pro-apoptotic shift in the BCLxL:BIM ratio.

	SYO-1	Aska-SS	Yamato-SS	HS-SY-II
IC 50 to ABBV-075	0.7 nM	4.8 nM	30 nM	67 nM
rate of change	76.8	43.19	5.10	6.02

## Data Availability

The datasets generated and/or analyzed during this research are contained within the article and Appendix A.

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
