# Peer review of "Therapeutic Potential of Bromodomain and Extra-Terminal Domain Inhibitors for Synovial Sarcoma Cells"

_cancers, 2024, doi:10.3390/cancers16061125_

Round 1

Reviewer 1 Report

Comments and Suggestions for Authors

The study examines bromodomain and extra-terminal domain (BET) inhibitors, which interfere with histone modification readers, known as bromodomains. By disrupting bromodomain binding, BET inhibitors reduce expression of cancer-associated genes, leading to cell cycle arrest and apoptosis (cell death). The study finds that BET inhibitors affect cell cycle regulators and BCL2 family proteins in synovial sarcoma cells. Additionally, blocking SS18-SSX reduces BCL2 expression and decreases sensitivity to BET inhibitors. These results suggest that BET inhibitors could be a promising therapy for synovial sarcoma by targeting the intrinsic apoptotic pathway modulated by SS18-SSX.

I recommend publishing it as it is.

Reviewer 2 Report

Comments and Suggestions for Authors

In the present study, the authors reported the potential of BET inhibitors for treating Synovial Sarcoma. The finding of this study is meaningful. However, there are other major limitations in the manuscript listed as followed.

1.         How to interpret the result of cell viability curve? First, the names of the compounds need to be labeled in the figures. Second, in Figure 1A, under 0.5uM, the response of NHDF is similar like the SS cells. And NHDF also responds to other BET inhibitors, will this limit the application of BET inhibitors for treating Synovial Sarcoma in clinic?

2.         What are the IC50s of NHDF for different BET inhibitors? These data need to be added in table 1.

3.         How the clinical usage or the development status of BET inhibitors in clinic?

4.         Is fusion gene SS18-SSX exist in 100% Synovial Sarcoma patients? Is it the oncogene for the occurrence of SS? Are the cell lines used in the article containing this oncogene?

5.         Except the apoptosis gene changes, what about the cell viability changes of the SS cells when knock down SS18-SSX upon BET inhibitors.

Comments on the Quality of English Language

Minor mistakes found.

Reviewer 3 Report

Comments and Suggestions for Authors

The manuscript entitled “Therapeutic Potential of BET Inhibitors for Synovial Sarcoma Cells” aimed to investigate the antitumor response of BET Inhibitors in an aggressive subset of sarcoma. Please, see my specific comments below:

1.     The introduction section is too short and does not provide sufficient background to understand the manuscript’s goals. Please increase readability.

2.     Please provide the validation statement for the cell lines. Were the cells subjected to validation and proof of human origin and have no contaminating cells? How about a mycoplasma test?

3.     It is unclear why western blot and qPCR were performed in the methodology. Was compared non-treated and treated cells? Please, provide one phrase with background to justify.

4.     I lacked an Ethics statement in this manuscript. Since the authors used a Xenograft model, it is mandatory. This brings serious concern regarding manuscript confirmability.

5.     The immunohistochemistry description did not allow replication. Are missing information. Please provide a complete description or a reference to the previous methodology.

6.     The results and conclusion present a good quality.

Round 2

Reviewer 2 Report

Comments and Suggestions for Authors

No further comments about the revised manuscript and can be accepted for publication with the current version.